# Association of Genetic Polymorphisms with Complications of Implanted LVAD Devices in Patients with Congestive Heart Failure: A Kazakhstani Study

**DOI:** 10.3390/jpm12050744

**Published:** 2022-05-04

**Authors:** Madina R. Zhalbinova, Saule E. Rakhimova, Ulan A. Kozhamkulov, Gulbanu A. Akilzhanova, Galina K. Kaussova, Kenes R. Akilzhanov, Yuriy V. Pya, Joseph H. Lee, Makhabbat S. Bekbossynova, Ainur R. Akilzhanova

**Affiliations:** 1National Laboratory Astana, Nazarbayev University, Nur-Sultan 010000, Kazakhstan; madina.zhalbinova@nu.edu.kz (M.R.Z.); saule.rakhimova@nu.edu.kz (S.E.R.); ulan.kozhamkulov@nu.edu.kz (U.A.K.); 2Department of General Biology and Genomics, L. N. Gumilyov Eurasian National University, Nur-Sultan 010000, Kazakhstan; 3Semey Medical University, Pavlodar Branch, Pavlodar 140000, Kazakhstan; gulbanu-25@mail.ru (G.A.A.); a_kenes79@mail.ru (K.R.A.); 4Kazakhstan Medical University “KSPH”, Almaty 050000, Kazakhstan; g.kausova@mail.ru; 5National Research Cardiac Surgery Center, Nur-Sultan 010000, Kazakhstan; pya_yuriy@mail.ru (Y.V.P.); mcardio_s@mail.ru (M.S.B.); 6Sergievsky Center, Taub Institute, Columbia University Irving Medical Centerx, 630 W, New York, NY 10032, USA; jhl2@cumc.columbia.edu

**Keywords:** genotype, polymorphism, heart failure, left ventricular assist device (LVAD), personalized medicine, thrombosis, bleeding

## Abstract

The left ventricular assist device (LVAD) is one of the alternative treatments for heart failure (HF) patients. However, LVAD support is followed by thrombosis, and bleeding complications which are caused by high non-physiologic shear stress and antithrombotic/anticoagulant therapy. A high risk of complications occurs in the presence of the genotype polymorphisms which are involved in the coagulation system, hemostasis function and in the metabolism of the therapy. The aim of the study was to investigate the influence of single-nucleotide polymorphisms (SNP) in HF patients with LVAD complications. We analyzed 21 SNPs in HF patients (*n* = 98) with/without complications, and healthy controls (*n* = 95). SNPs rs9934438; rs9923231 in *VKORC1*, rs5918 in *ITGB3* and rs2070959 in *UGT1A6* demonstrated significant association with HF patients’ complications (OR (95% CI): 3.96 (1.42–11.02), *p* = 0.0057), (OR (95% CI): 3.55 (1.28–9.86), *p* = 0.011), (OR (95% CI): 5.37 (1.79–16.16), *p* = 0.0056) and OR (95% CI): 4.40 (1.06–18.20), *p* = 0.044]. Genotype polymorphisms could help to predict complications at pre- and post-LVAD implantation period, which will reduce mortality rate. Our research showed that patients can receive treatment with warfarin and aspirin with a personalized dosage and LVAD complications can be predicted by reference to their genotype polymorphisms in *VKORC1*, *ITGB3* and *UGT1A6* genes.

## 1. Introduction

Implantation of a left ventricular assist device (LVAD) is one of the best alternative treatments for end-stage heart failure (HF) patients, instead of heart transplantation (HT) [1]. After LVAD implantation patients experience improvements in their quality of life with reduced mortality. The survival rate after HT is 86%, while survival rate after LVAD implantation is 80%, which represents an optimal outcome [2]. In addition, LVAD implantation is better choice considering the acute shortage of the heart donors. Patients on the waiting list for HT have an increased risk of mortality rate compared to those patients who receive LVAD [3].

LVAD is implanted as a bridge-to-transplantation (BTT) or a destination therapy (DT). BTT is prescribed as a temporary treatment for patients who are heart donor candidates. DT is implanted as a lifetime support for patients who are not able to have HT because of their age, medical indications and complications [2,4,5,6].

However, LVAD is followed by complications which cause harm to quality of life in the first year after implantation. Bleeding and thrombosis are the most common complications, which require re-hospitalization [2,7,8,9,10]. Fifty percent of the mortality is caused by pump thrombosis and stroke. Consequently, patients require pump exchange or HT [8]. Anticoagulation therapy such as warfarin or aspirin is usually prescribed to reduce the risk of thromboembolic complications and pump thrombosis. However, a high dose of anticoagulation therapy predisposes patients to bleeding risks with a mortality rate of 9–10% [2,8,11,12]. Nowadays, complications can be reduced by genotype-guided warfarin therapy according to the genotyping test results for genetic polymorphisms of *VKORC1* (vitamin K epoxide reductase complex 1) and *CYP2C9* (cytochrome P450 2C9) genes. These genes are involved in warfarin metabolism and action [9,11,13]. Genome-wide association studies (GWAS) have shown that *VKORC1* and *CYP2C9* genes demonstrated 30% variability after a warfarin dose in European and Asian populations in non-LVAD patients [13,14,15]. Research on HF patients with implanted LVAD showed a difference in genotype frequency compared to other races such as European–American and African–American. Frequency of mutant variants for *CYP2C9* and *VKORC1* genes were significantly higher in European–Americans (38.0% and 50%, *p* < 0.05) than in African–Americans (9.7% and 3.2%, *p* < 0.05) [9]. There were no other investigations on GWAS in HF patients with implanted LVAD and their associations with complications [9,11,13].

One common reason for the blood clotting and bleeding events is platelet dysfunction [16]. Platelet dysfunctions are caused by the high non-physiologic shear stress (NPSS) at the blade region of the LVAD’s rotary. Shear stress induces platelet receptor damage and shedding. There are three important adhesive glycoprotein receptors on the membrane of the platelet: GPIbα, GPVI, and GPIIb/IIIa. Platelet activation, aggregation and adhesion processes occur when glycoprotein receptors GPIbα bind to von Willebrand factor (VWF), GPVI binds to collagen, and the GPIIb/IIIa to fibrinogen binds to VWF [17].

Apart from the shear-stresses, patient’s genetic information also influences the development of LVAD complications. Consequently, the purpose of our research is to study the influence of single-nucleotide polymorphisms (SNP) of the genes encoding blood coagulation processes, the metabolism of the anticoagulant, on antithrombotic therapy and the impact on complications development after LVAD implantation in HF patients.

## 2. Materials and Methods

### 2.1. Study Participants

The study was conducted in accordance with the Declaration of Helsinki. Research protocol was approved by the Ethics Committees of the National Laboratory Astana, Nazarbayev University (No.16 from 11 March 2015) and the National Research Cardiac Surgery Center (NRCC), Nur-Sultan, Kazakhstan (No.16 from 24 April 2015). Written informed consent was obtained from all participants. During the data collection, all personal information on patients is encoded and data depersonalized, to protect the rights of patients and not disclose their personal information. Written informed consent has been obtained from the patient(s) to publish this paper.

We recruited consecutively 100 HF patients with implanted LVAD during 2011–2016 when they were implanted with LVAD as BTT and DT at the NRCC. Two patients under 18 years old (9 and 16 years old) were not included in the analysis. HF patients (age ≥ 18) were diagnosed with ischemic cardiomyopathy (ICM), dilated cardiomyopathy (DCM), hypertrophic cardiomyopathy (HCM) and valvular heart disease (VHD) by the cardiologists at NRCC. New York Heart Association (NYHA) functional class IIIA was noted in 34 patients (34.7%), class IIIB in 34 patients (34.7%) and class IV in 26 patients (26.5%) before LVAD implantation (Table 1). Ninety-seven (99%) patients were suffering from heart failure reduced ejection fraction (HFrEF). There were three types of LVADs implanted: 18 (18.4%) HeartWare HVADs (HW) (HeartWare Inc., Framingham MA, USA), 34 (34.7%) HeartMate II (HM2) (Thoratec Corporation, Pleasanton, CA, USA) and 46 (46.9%) HeartMate III (HM3) (St Jude Medical, Huntingdon, Cambridgeshire, UK) (Table 1). Patients had three-years’ follow up from 2014 till 2017 with a median of 18 months. Venous blood samples were obtained in sterile vacutainers with K2EDTA for further genetic analysis.

After LVAD implantation, warfarin and aspirin were prescribed according to the clinical protocol of the Ministry of Healthcare of the Republic of Kazakhstan. Warfarin dose was corrected to maintain an international normalized ratio range (INR 2.25–3.25). Twenty-four (24.5%) patients suffered one of the post-implantation complications, such as thrombosis, bleeding and infections. Three patients had both thrombosis and bleeding complications.

Patients were categorized into two groups for comparative analysis: Group 1 (*n* = 74, without complications), and Group 2 (*n* = 24, with complications). Baseline demographic characteristics were compared between these two groups (Table 2); biochemical parameters also were compared between these two groups (Appendix A).

For more detailed analysis, the group of patients with complications (Group 2) was classified into two subgroups: with thrombosis (Group 2-1) and bleeding (Group 2-2). Baseline demographic characteristics were analyzed comparing these two subgroup of patients with complications (Appendix A). Comparison of biochemical parameters was further analyzed comparing Group 1 (without complications) and subgroup 2-1 (thrombosis), and subgroup 2-2 (bleeding), respectively (Table 3).

Ninety-five healthy individuals without any cardiovascular diseases at the time of recruitment (63 males and 32 females, 44.01 ± 13.8 years old) were included in the study as a control group for genetic analysis, representing the general population.

Clinical and epidemiological data were retrieved from medical records of patients at NRCC. Data included demographic parameters (age, gender, ethnicity), anthropometry (height, weight, body mass index (BMI)), LVAD type, systolic blood pressure (SBP), diastolic blood pressure (DBP), device type, echocardiography, thrombo-elastography (TEG), clinical biochemistry, including coagulation factors, lactase dehydrogenase (LDH), lupus anticoagulant (LA), and others.

Furthermore, we decided to compare patients without complications (Group 1) with thrombosis (Group 2-1) and bleeding (Group 2-2) subgroups, respectively (Table 3). Patients with bleeding events had significantly lower level of hemoglobin, hematocrit, lymphocytes and AST than Group 1 (*p* < 0.05). In addition, patients with thrombosis had significantly different values of lymphocytes, LDH and total bilirubin than those in Group 1 (*p* < 0.05). Other biochemical parameters did not show any statistically significant difference on the occurrence of complications (Appendix A).

### 2.2. Selection of the SNPs

Twenty-one SNPs in *VKORC1* (rs8050894, rs9934438, rs9923231), *CYP2C9* (rs1799853, rs1057910, rs28371686), *CYP2C19* (rs4244285, rs4986893), *ITGB3* (rs5918), *GGCX* (rs11676382), *CYP4F2* (rs2108622), *UGT1A6* (rs2070959), *ACSM2A* (rs1133607), *PTGS1* (rs3842787), *F5* (rs6025), *F13A1* (rs5985), *F2* (rs1799963), *F7* (rs6046), *FGB* (rs1800790), *MTHFR* (rs1801133, rs1801131) genes encoding coagulation system, metabolism of warfarin and aspirin, and previously showing association with cardiovascular events, were selected in this study [9,18,19,20]. We analysed known polymorphisms of *VKORC1*, *CYP2C9, GGCX* and *CYP4F2* genes which are involved in warfarin metabolism and studied in HF patients with warfarin dosing in anticoagulant treatment [9,11,13,21,22,23]. Polymorphisms in genes *CYP2C19, ITGB3, UGT1A6,*
*ACSM2A, PTGS1* were included in the research based on their association with the antiplatelet effect of aspirin in cardiovascular events [17,18,24,25]. Genes of coagulation factors such as *F5, F13A1, F2, F7, FGB* and *MTHFR* involved in the folate/homocysteine metabolism pathway were selected due to their associations with thrombosis events [25,26,27]. A list of SNPs and used primers are summarized in Appendix A.

### 2.3. DNA Extraction and SNP Genotyping

DNA extraction and genotyping were processed at the National Laboratory Astana, Nazarbayev University, Nur-Sultan, Kazakhstan. Genomic DNA was extracted from 200 μL whole blood samples using the PureLinkTM Genomic DNA Mini Kit (Invitrogen, Carlsbad, CA, USA) according to the manufacturer’s instructions. The concentration and purity of the extracted DNA were measured by NanoDrop™ Spectrophotometer (ThermoFisher Scientific, Waltham, MA, USA). Further, DNA samples were genotyped by using real-time polymerase chain reaction (qPCR) with allele discrimination using TaqMan Real Time PCR Assay on a 7900HT Fast Real-Time PCR System (Applied Biosystems, Waltham, MA, USA). qPCR mixture contained Universal Master Mix (2×) (Applied Biosystems, Waltham, MA, USA), 0.25 µL of TaqMan probe (40×) respectively (Applied Biosystems, Waltham, MA, USA) and 10 ng of DNA. Reaction volume was achieved with mqH_2_O. Further, qPCR results were determined by using SDS version 2.4 software.

### 2.4. Statistical Analysis

Continuous variables were presented as mean ± standard deviation (SD). Categorical variables were performed as percentages (%) and compared using chi-square test or Fisher’s exact test. Normality of distribution of continuous variables was assessed by using a Kolmogorov-Smirnov test (*p* > 0.05). Continuous variables were compared between two groups by Student’s *t*-test (normally distributed variables) and by non-parametric Mann—Whitney U test (not normally distributed variables). Sample size calculation and power analysis were identified by online calculator on https://clincalc.com, accessed on 5 september 2021. Sample sizes in the group of 75.5% (*n* = 74, without complications) and 24.5% (*n* = 24, with complications) achieved 0.85 (85%) power, with alpha value of 0.05. Each group needs at least 16 patients according to the sample size calculation results.

Hardy-Weinberg equilibrium (HWE) for genetic deviation was performed using chi square test or Fisher’s exact test. Associations between polymorphisms and HF patients with/without complications were evaluated by using odds ratios (OR) with 95% confidence interval (CI) and *p* value. Logistic regression analysis was performed using the web tool https://snpstats.net/, accessed on 9 March 2022. Bonferroni adjusted *p*-values were calculated for association of SNPs between multiple comparisons of healthy control group and HF patients with/without complications. A one-way ANOVA was done between genotype groups of statistically significant SNP genotypes. *p*-value < 0.05 was considered statistically significant. The statistical analysis was performed in SPSS program version 23 (SPSS, Chicago, IL, USA) Gene–gene interactions of all 21 SNPs in HF patients were investigated by Multifactor Dimensionality Reduction (MDR) analysis. MDR analysis was used for identification of low-risk and high-risk of genotypes. MDR analysis for 2 groups (with/without complications) of HF patients was assessed via R package. This package assumes binary case-control data with categorical predictor variables [28,29]. The binary response variable was coded as 0 or 1, and the categorical predictors (SNP genotypes) were coded numerically (0, 1, 2). The best MDR model was chosen by using a three-way split as internal validation methods to prevent over-fitting [29].

Furthermore, we studied the influence of genotype polymorphisms in the prediction of LVAD complications between HF patients (with/without complications) by using ROC analysis. ROC analysis was performed by using G-WIZ program in R [30]. To perform analysis by G-WIZ, we created file (csv.) with the study sample size, OR and risk allele frequencies for each SNP in the study.

## 3. Results

### 3.1. Clinical Characteristics of HF Patients with and without Complications

Baseline demographic characteristics of control group, HF patients and HF patients with and without complications are shown in Table 1 and Table 2. Among the 98 cases, 44 patients were diagnosed with ICM (44.9%) and 40 with DCM (40.8%) (Table 1). In our patient cohort, complications were more prevalent in patients with implanted HM2 device in 12 (50%) cases, and in patients with HW and HM3 in 7 cases (29.2) and 5 cases (20.8), respectively (*p* = 0.01) (Table 2). Most of the patients had implantation of LVAD as DT (89.8%), because heart donors were not available for every patient during the follow up period (Table 1). After implantation of the LVAD, 13 cases (54.2%) of thrombosis, 14 cases (58.3%) of bleeding and 15 (62.5%) of infection were developed in the group of patients with complications (*p =* 0.0001, *p* = 0.0001 and *p* = 0.015 for each type of complication respectively) (Table 2). The initial mean warfarin dose was 2.99 ± 1.15 mg/day and aspirin dose was prescribed a1s 00 mg daily.

In the period of this investigation (2011–2016), the mean duration of LVAD support was 29.6 ± 17.3 months. Seventy-one (72.4%) patients reached outcome measurement time, and among them, 10 (10.2%) patients had heart transplantation (BTT) (Table 1). Twenty-seven (27.6) patients died before reaching the time point of the outcome measurement (Table 1). In the group of patients who achieved outcome measurement, 58 (78.4%) had no complications (Group 1) and complications developed in 13 (54.2%) cases (Group 2), whereas among deceased patients complications developed in 11 cases (45.8%) (*p* = 0.03) (Table 2).

Biochemical parameters of pre- and post-LVAD implantation data were compared between patients with/without complications (Appendix A). Hematocrit level after 6 months of LVAD implantation was significantly lower in the group of patients with complications than in those without complications (*p* = 0.047). According to the Mann-Whitney U test biochemical parameters such as lymphocytes, AST and total bilirubin level were significantly higher in the group of patients with complications (*p* < 0.05). The rest of the biochemical parameters did not show an effect on the development of complications (Appendix A).

### 3.2. Analysis of Genotyping

Twenty-one SNPs were genotyped in HF patients (*n* = 98) with implanted LVAD and healthy controls (*n* = 95). The distributions of allelic and genotype frequencies among HF patients and controls, and among HF patients with/without complications, are summarized in Table 4 and Table 5. The distributions of allelic frequencies were tested for Hardy-Weinberg equilibrium (HWE).

The distributions of allelic and genotype frequencies of two SNPs rs8050894 in the *VKORC1* and rs5918 in the *ITGB3* genes were significantly different between HF patients and healthy control groups (*p* < 0.05) (Table 4). CC genotype of rs8050894 polymorphism in *VKORC1* C > G gene was significantly higher in HF patients than in the control group (56.1% vs. 17.9%, *p* = 0.0001). CC genotype of rs5918 polymorphism in *ITGB3* T > C gene was also significantly higher in HF patients than in control group (16.3% vs.1.1%, *p* = 0.0001). Nineteen SNPs were not significantly different between HF patients and healthy control groups (*p* > 0.05). Distribution of genotypes and alleles of studied polymorphisms in our study correspond to the population level according to 1000 Genomes, Ensemble, and NCBI databases. Allele frequency levels of 21 polymorphisms in HF patients were more common to Asian and East Asian populations. Results of allele frequency of HF patients, healthy control group and other different populations are summarized in Appendix A.

Furthermore, we found that the distributions of allelic and genotype frequencies of SNPs rs9934438, rs9923231 in the *VKORC1* gene, rs5918 in the *ITGB3* gene and rs2070959 in the *UGT1A6* gene were significantly different between HF patient groups with and without complications (Table 5). Carriers of the GA genotype for polymorphism rs9934438 in *VKORC1* G > A gene and CT genotype for rs9923231 were significantly higher in the group of patients with complications than in those patients without complications (70.8 vs. 39.2 and 70.8% vs. 41.9%, *p* < 0.05). TC genotype of rs5918 in ITGB3 T > C gene was significantly higher in the group of patients with complications than that in patients without complications (62.5% vs. 25.7%, *p* = 0.005). GG genotype of rs2070959 in *UGT1A6* A > G gene was significantly higher in the group of patients with complications than in patients without complications (20.8% vs. 6.8%, *p* = 0.03).

Logistic regression analysis (adjusted for age, BMI and gender) showed that SNPs rs9934438; rs9923231 in *VKORC1*, rs5918 in *ITGB3* and rs2070959 in *UGT1A6* are significantly associated with patients with complications (Table 6). GA genotype of rs9934438 and CT genotype of rs9923231 in *VKORC1* gene showed significant association with complications [OR (95% CI): 3.96 (1.42–11.02), *p* = 0.0057 and (OR (95% CI): 3.55 (1.28–9.86), *p* = 0.011]) TC genotype of rs5918 in *ITGB3* gene and GG genotype of rs2070959 in the *UGT1A6* gene were also significantly associated with the group of patients with complications [(R (95% CI): 5.37 (1.79–16.16), *p* = 0.0056 and OR (95% CI): 4.40 (1.06–18.20), *p* = 0.044]. Logistic regression analysis results for SNPs are summarized in Appendix A.

Polymorphisms of rs8050894 in *VKORC1* and rs5918 in *ITGB3* genes were found to be significantly associated with HF and complications according to the Bonferroni correction analysis (*p* < 0.001).

We performed MDR analysis to identify gene–gene interactions of 21 SNPs in the HF patients. We considered all combinations of 21 SNPs up to size K = 3 and default settings for the other options. The best MDR model combinations according to the three-way split internal validation are summarized in Table 7.

MDR fit identified that combinations of polymorphisms rs9934438 in *VKORC1*2*, rs2070959 in *UGT1A6* and 1801133 in *MTHFR*1* genes are the best model of disease prediction status (in our case, whether HF patients with/without complications) with prediction accuracy of 95.83% (Table 7).

We can visually display the results for MDR fit and see which genotype combinations are at high-risk. In our case, eight combinations of polymorphisms rs9934438 in *VKORC1*2* gene, rs2070959 in *UGT1A6* gene and rs1801133 in *MTHFR*1* gene are classified as high-risk for complications in HF patients (Figure 1).

Further, we repeated analysis with combination size for K = 4 (Table 8). This includes combinations of polymorphisms rs5918 in *ITGB3* gene, rs2108622 in *CYP4F2* gene, rs1801133 in *MTHFR*1* gene and rs1800790 in *FGB* gene, which provides gave the best predictive status (96.77%) as to whether patients will experience complications or not. The plot of MDR fit was also extracted, but due to the large number of combinations the plot is presented in a Appendix A.

ROC analysis included the study sample size, OR, risk allele frequencies and model type for each SNP of the study (Appendix A). ROC analysis was performed for 19 SNPs except polymorphisms of rs28371686 in *CYP2C9*5* gene and rs1799963 in *F2* gene. HF patients were carriers only of wild genotypes for rs28371686 in *CYP2C9*5* and rs1799963 in *F2* genes. The ROC curve for 19 SNPs is plotted in Figure 2A and the area under the ROC curve was 0.85 which showed a good discriminator between HF patients with and without complications. Based on this ROC curve and AUC score, we could claim that 19 SNPs have good predictive value on HF patients’ complications status.

We also performed ROC analysis for four SNPs rs9934438, rs9923231 in *VKORC1* gene, rs5918 in *ITGB3* gene and rs2070959 in *UGT1A6* gene, which were significantly associated with HF patients’ complications according to the logistic regression analysis (Table 6). The ROC curve is demonstrated in Figure 2B. AUC score for 4 SNPs was 0.68 which was lower in comparison to the ROC curve for 19 SNPs.

Significantly different SNPs rs8050894, rs9934438, rs9923231 in *VKORC1*, rs5918 in *ITGB3* and rs2070959 in *UGT1A6* were analyzed in HF patients with and without complications. We performed analyses of biochemical parameter levels according to genotypes of SNPs rs8050894, rs9934438, rs9923231 in *VKORC1*, rs5918 in *ITGB3* and rs2070959 in *UGT1A6* respectively in patients with- and without- complications (Appendix A). Biochemical parameters such as hemoglobin, hematocrit, leukocytes, lymphocytes, erythrocytes, prothrombin time, aspartate aminotransferase (AST), alanine aminotransferase (ALT), total bilirubin, blood urea nitrogen (BUN) and other characteristics were significantly associated with three SNPs (rs8050894, rs9934438, rs9923231) in *VKORC1*, one SNP (rs5918) in *ITGB3* and one SNP (rs2070959) in *UGT1A6* genes (*p* < 0.05).

## 4. Discussion

This study aimed to investigate the genotype distributions of 21 SNPs which are involved in the coagulation system, warfarin and aspirin metabolism in HF patients with/without complications due to LVAD implantation, and in a healthy control group. We selected SNPs which were studied previously in HF patients and in other cardiovascular events. Polymorphisms of VKORC1 and CYP2C9 genes were studied previously in LVAD patients [9,11]. However, we did not find in the available literature that other SNPs were investigated specifically in HF patients with implanted LVAD [23,25]. Distribution of genotypes and alleles of all 21 SNPs in our study correspond to the population level according to 1000 Genomes, Ensemble, and NCBI databases, which showed a similar frequency to Asian and East Asian populations (Appendix A).

Out of 21 SNPs, distributions of the allelic and genotypic frequencies of polymorphisms rs8050894 in *VKORC1* gene and rs5918 in *ITGB3* gene were significantly different between HF patients and healthy controls (Bonferroni correction analysis (*p* < 0.001)). In addition, we found significant differences in distributions of the allelic and genotypic frequencies of polymorphisms rs9934438, and rs9923231 in *VKORC1* gene, rs5918 in *ITGB3* gene and rs2070959 in *UGT1A6* gene between HF patients with and without complications. Logistic regression analysis revealed significant association of rs9934438; rs9923231 in *VKORC1*, rs5918 in *ITGB3* and rs2070959 in *UGT1A6* with complications of HF patients. These SNPs were further analyzed in detail for identification their genetic influence on HF patients with/without- complications.

We performed MDR analysis to identify gene–gene interactions of 21 SNPs in the HF patients. MDR fit identified that combinations of polymorphisms rs9934438 in *VKORC1*2*, rs2070959 in *UGT1A6* and 1801133 in *MTHFR*1* genes are the best model of disease prediction status (in our case, whether HF patients were with/without complications) with prediction accuracy of 95.83% (Table 7).

Warfarin’s metabolism and dose effects depend on genetic variants of vitamin K epoxide reductase complex subunit 1 (*VKORC1*) [1,9,11,19]. Genotypic polymorphisms of VKORC1 gene could help to prevent over- and under-coagulation by predicting what warfarin dosage is optimal, which can reduce thrombosis and bleeding complications [9,11].

*VKORC1* polymorphisms have different genotypic distributions across different racial groups such as African-American, European-American, Asian and Caucasian [9,11,20,21]. Scott et al. identified significantly different allelic frequency of polymorphism rs9923231 in *VKORC1* gene between African-American and Asian populations (*p* < 0.0001). Investigation revealed that an Asian population had significantly higher frequency of mutant allele (-1639A) of polymorphism rs9923231 in *VKORC1* gene than wild type allele (-1639G) [20]. Our study also showed that healthy control group and HF patients have higher frequency of mutant T allele of polymorphism rs9923231 in *VKORC1* gene similar to Asian population (Appendix A). Moreover heterozygote genotype polymorphisms for rs9934438 and rs9923231 in *VKORC1*gene are significantly associated with the group of HF patients with complications (*p* < 0.05).

A higher warfarin dose is recommended with the presence of the wild type genotypic polymorphisms in *VKORC1* gene [9,11]. On the other hand, lower warfarin dose is usually prescribed with the presence of the mutant genotype polymorphisms [9,19,22,31]. Our research found significant difference of prescribed warfarin dosage in the clinic between genotype polymorphisms of rs8050894, rs9934438, rs9923231 in *VKORC1* gene among HF patients with LVAD implantation (*p* < 0.05) (Appendix A). HF patients with the presence of the wild type GG genotype and heterozygote GA genotype for rs9934438 in *VKORC1* gene were prescribed with higher warfarin dosage with maximum of 3.88 ± 1.25 mg/day. HF patients; carriers of mutant AA genotype for rs9934438 were prescribed with lower warfarin dose with maximum of 2.44 ± 0.81 mg/day. However, in our research we did not find any significant association of genotype polymorphisms in *VKORC1* gene separately with thrombosis and bleeding complication groups (*p* < 0.05). On the contrary, Topkara et al. (2015) found that HF patients had significantly higher risk of thrombosis complications with mutant genotype polymorphism of rs9923231 in *VKORC1* gene than with wild type genotype. However, there was no association with gastrointestinal bleeding complications in HF patients with implanted LVAD [32]. Warfarin dose difference for each genotype polymorphisms rs8050894, rs9934438, rs9923231 in *VKORC1* gene for HF patients is summarized in Appendix A. Our result supported that genotype guided warfarin dosage before prescription will reduce both mortality and development of complications in HF patients (thrombosis, bleeding) after LVAD implantation, consistent with previous reports [9,11].

LVAD’s shear stress causes platelet dysfunction which affects normal hemostatic function. Investigations revealed that high non-physiologic shear stress (NPSS) causes loss of the GPIbα platelet glycoprotein receptor which affects increased bleeding events in HF patients with implanted LVAD [33]. NPSS also causes thrombosis complications by enhanced activation of GPIIb/IIIa receptors [14,34]. If NPSS causes damage of the platelet receptors, the genetic polymorphism of the receptors also needs to be considered [35]. Platelet’s function may vary by 30% due to genetic diversity. Genetic polymorphisms of platelet receptors have associations with high risk of bleeding and thrombosis events [17,35,36]. Genetic factors of glycoprotein GPIIb/IIIa receptors, fibrinogen, prothrombin, VWF, factor V, factor VII, factor XIII and plasminogen activator inhibitor-1 have major role in the process of hemostasis and are associated with side effects in cardiovascular events. Consequently, these genetic factors influence the treatment outcome in cardiovascular diseases [16,17,18,35,37,38].

This current investigation also included the polymorphism rs5918 of *ITGB3* gene which encodes platelet receptor GPIIIa [17,38]. There are reports that mutant CC genotype of rs5918 in the *ITGB3* gene has higher risk of thrombosis formation in cases of stroke, coronary artery disease and myocardial infarction (MI) [18,38,39,40,41]. Our study showed that only heterozygote TC genotype of rs5918 in ITGB3 gene was significantly associated with HF patients who have LVAD complications such as thrombosis and bleeding (*n* = 24) (OR (95% CI): 5.37 (1.79–16.16), *p* = 0.0056). Apart from the side effects of NPSS, platelet receptor GPIIIa with TC genotype of rs5918 in *ITGB3* gene could have heritable platelet dysfunction which causes higher risk of complications in HF patients with implanted LVAD. Previous investigations did not consider genes encoding platelet receptors for the predictions of the complications in HF patients with implanted LVAD. Identification of the warfarin dose by polymorphisms of *VKORC1* and *CYP2C9* genes is not enough for the prediction and prevention of complications [9,11]

Acetyl salicylic acid (aspirin) is one of the most effective nonsteroidal anti-inflammatory drugs (NSAID), which deacetylates to salicylic acid after absorption. The metabolism of aspirin occurs in two phases: (1) Phase-I by enzyme of cytochrome P450s; (2) Phase-II by glucuronidation which is catalyzed by UDP-glucuronosyltransferases (UGTs) [24,42,43]. *UGT1A6* is a major enzyme involved in aspirin metabolism. Genetic polymorphism in *UGT1A6* can influence to the expression of different metabolic activities of the enzyme [24,43,44]. *UGT1A6* gene polymorphism rs2070959 was found to be associated with colon and colorectal cancer [24,45,46,47]. Furthermore, *UGT1A6* gene polymorphism rs2070959 was found not to be associated in cardiovascular patients with gastrointestinal complaints in aspirin treatment [42].

Aspirin metabolites excrete faster in individuals with GG genotype of *UGT1A6* rs2070959, and on the contrary, *UGT1A6* rs2070959 AA genotype carriers excrete a smaller amount of aspirin metabolites because of slower metabolism [44,45]. Patients with GG genotype require a higher aspirin dose due to the enzyme’s higher metabolic activity [44]. However, there were controversial findings regarding this SNP, showing that GG genotype polymorphism rs2070959 in *UGT1A6* gene is associated with decreased enzyme activity [46,48].

In our study, distribution of mutant GG genotype of rs2070959 in the *UGT1A* was significantly associated with HF patients with complications (OR (95% CI): 2.67 (0.59–12.07), *p* = 0.03). Distribution of AA genotype of rs2070959 in the *UGT1A6* gene was higher in HF patients with complications than those patients without complications (50% vs. 37.8%). Patients (*n* = 24) had development of complications such as pump thrombosis, stroke, and internal jugular vein thrombosis, on average after 24.5 months of LVAD implantation (from 7 months to 54 months). There were also complications in the form of bleeding from the gastrointestinal tract, gynecological, nasal, rectal bleeding and gingival bleeding, which developed on average 9.4 months after implantation (from 1 month to 24 months). These complications in several patients were even diagnosed twice after the implantation of the device.

According to the HF patients’ listed complications and their genotypes, our research found that 100mg of aspirin was not enough for patients with GG genotype of rs2070959 in the *UGT1A6* gene, which excretes faster aspirin metabolites. On the contrast, the same amount of aspirin was high for patients with AA genotype of rs2070959 in the *UGT1A6* gene, which is responsible for the slower excretion of aspirin metabolites. Identification of genotypes of rs2070959 in *UGT1A6* gene could help to guide the prescription of aspirin individual dosage for reduction of complications before and after LVAD implantation. Various UGT polymorphisms should be studied for the identification of patient’s precise aspirin dosage, which will help to reduce complications and mortality rate [43]. Polymorphism rs2070959 in *UGT1A6* gene is the first polymorphism studied in HF patients with thrombosis and bleeding complications after implanted LVAD devices [42,45,47,49]. LVAD patients should be prescribed with individualized aspirin dosage according to the results of genotyping for the polymorphisms of *UGT1A6* gene.

We also analyzed biochemical parameters between control, thrombosis and bleeding subgroups (Appendix A). Patients with bleeding events had significant differences in hemoglobin, hematocrit and lymphocyte levels compared to patients without complications (*p* < 0.05). Significant reduction of hemoglobin and hematocrit levels were identified in patients with bleeding events [7,50]. In this study, we showed that HF patients with bleeding complications had significantly lower level of hemoglobin (111.1 ± 29.2 vs. 129.5 ± 16.2, *p* = 0.012) and lower level of hematocrit (32.8 ± 8.70 vs. 38.2 ± 4.92, *p* = 0.016) than those HF patients without complications at post-LVAD (*p* < 0.05). On the other hand, HF patients with thrombosis complications had significantly different levels of lymphocytes and total bilirubin (*p* < 0.05). There are investigations showing that a lower level of hematocrit could be a predictor of bleeding events pre-LVAD [50]. Our results showed that lower levels of hemoglobin and hematocrit could be a predictor of bleeding complications at both pre- and post-LVAD implantation period. Biochemical parameters can predict development of complications before LVAD implantation and such biochemical measurements might help to reduce mortality rate among HF patients.

Furthermore, we compared biochemical parameters between carriers of different genotypes of polymorphisms rs8050894, rs9934438, rs9923231 in *VKORC1* gene, rs5918 in *ITGB3* gene and rs2070959 in *UGT1A6* gene for identification of genetic influence among HF patients with/without complications (Appendix A). Level of fibrinogen, hemoglobin, hematocrit, blood urea nitrogen, leukocytes, lymphocytes, total bilirubin and other biochemical parameters were significantly associated with the genotypes of *VKORC1, ITGB3* and *UGT1A6* genes (*p* < 0.05). Previously, we mentioned that lower level of hemoglobin (111.1 ± 29.2, *p* = 0.012) and hematocrit (32.8 ± 8.70, *p* = 0.016) could be a predictor of the bleeding complications in HF patients after LVAD implantation (Table 3). On the other hand, levels of hemoglobin (103.7 ± 9.61, *p* = 0.01) and hematocrit (32.3 ± 4.73, *p* = 0.01) were significantly lower in GG genotype polymorphisms of rs8050894 in *VKORC1* gene (*p* < 0.05). Our study identified that patients with GG genotype of rs8050894 in *VKORC1* gene could have lower level of hematocrit and hemoglobin with risks of bleeding complications after LVAD implantation.

Prothrombin time was also significantly different in carriers of genotypes CC and TT rs5918 of *ITGB3* gene (38.4 ± 20.4 vs. 23.9 ± 5.96, *p* = 0.03, respectively). Platelet receptor subunits GPIIb/IIIa are encoded with *ITGA2B* (GPIIb) and *ITGB3* (GPIIIa) genes. *ITGB3* and *ITGA2B* gene polymorphisms are associated with increased platelet activity. Defect of the platelet receptors due to genetic inheritance can cause Glanzmann’s thrombasthenia, which is followed by disorders of platelet aggregation and increased bleeding events. Investigations showed that increased prothrombin time was shown to be associated with polymorphism rs70940817 of *ITGB3* gene [51]. In future perspectives, our investigation should include other polymorphisms of *ITGB3* gene and discover stronger associations with platelet dysfunctions in HF patients after LVAD implantation. Identification of different gene polymorphisms and biomarkers in LVAD patients could prevent LVAD complications and mortality and develop new pharmacological treatments for a better life for HF patients [52].

We observed that carriers of mutant GG genotype of rs2070959 in *UGT1A6* gene had significantly higher level of bilirubin than wild type AA carriers (2.70 ± 2.35 vs. 0.88 ± 0.57, respectively, *p* = 0.04). Oussalah et al. (2015) in their research showed that polymorphisms of *UGT1A6* gene were associated with serum bilirubin concentration [53]. Moreover, other studies demonstrate that *UGT1A1*28* polymorphism is associated with hyperbilirubinemia in newborns [54]. *UGT1A1*28* polymorphism was found to be the main cause of Gilbert syndrome which is identified with chronic hyperbilirubinemia [54,55]. Our findings showed that HF patients, carriers of mutant GG genotype, had a high bilirubin level, which will need more detailed analysis in future studies.. Other polymorphisms of the gene should be studied in HF patients with implanted LVAD to clarify the role of this gene and association with bilirubin level.

## 5. Limitations

Our study involved 98 HF patients with implanted LVADs. The small sample size limited the statistical analysis power in sub-groups analysis. The low sample size is problematic, especially when trying to create comprehensively adjusted models. Some of the biochemical parameters data were not longitudinally available in this current study due to patients’ death.

## 6. Conclusions

Polymorphisms of rs8050894, rs9934438, rs9923231 in *VKORC1* gene, rs5918 in *ITGB3* gene and rs2070959 in *UGT1A6* gene showed significant association in HF patients who have LVAD with thrombosis and bleeding complications. Genotyping of polymorphisms could help to predict complications at both pre- and post-LVAD implantation periods which will reduce morbidity and mortality rate. Our research showed that patients can receive personalized treatment with warfarin and/or aspirin dosage by reference to their genotypes in polymorphisms of *VKORC1* and *UGT1A6* genes.

We also found that HF patients with bleeding and thrombosis events have a significant difference in biochemical parameters at pre- and post-LVAD implantation period. Changes in biochemical parameters could predict complications after LVAD implantation. Biochemical parameters were significantly associated with genotypes of polymorphisms in *VKORC1, ITGB3* and *UGT1A6* genes.

This study for the first time investigated genetic polymorphisms of rs2070959 in *UGT1A6* gene in HF patients with implanted LVAD devices in Kazakhstan. We concluded that LVAD complications can be predicted by genotyping of polymorphisms of genes which are involved in the coagulation system, hemostasis function and in the metabolism of the antithrombotic/anticoagulant therapy. Recent development of genomic technologies and decreasing high throughput sequencing price per Gb allows more wide applications of sequencing technologies in clinical practice. Next-generation sequencing technologies such as whole exome and whole genome sequencing could be useful in diagnosis of HF patients with implanted LVAD, prediction of complication risks and prevention. Machine learning models in addition to traditional logistic regression and other forms of analysis could achieve better capabilities of detecting risk genetic risk factors contributing to the development of heart failure and complications of antithrombotic and anticoagulant therapies [56]. Using NGS technologies and machine learning models in future perspective would prevent patients experiencing a high risk of complications, reduce mortality rate and shorten treatment days in-hospital stay after LVAD implantation. Genomic personalized treatment guided by innovative technologies will bring great achievements in treatment of HF patients before prescription of LVAD implantation.

## Figures and Tables

**Figure 1 jpm-12-00744-f001:**
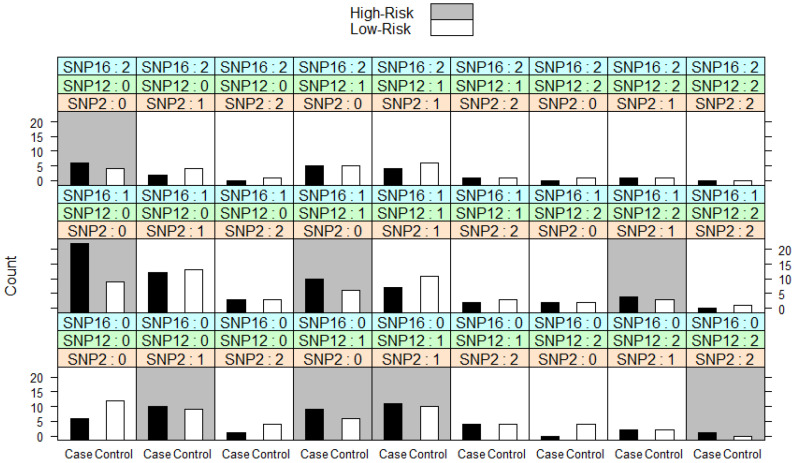
The illustration of MDR fit with three-way split on LVAD dataset with 98 individuals at 21 SNPs (k = 3). SNP 2 is polymorphism rs9934438 in *VKORC1*2* gene; SNP12 is polymorphism rs2070959 in *UGT1A6* gene; SNP16 is polymorphism rs1801133 in *MTHFR*1*.

**Figure 2 jpm-12-00744-f002:**
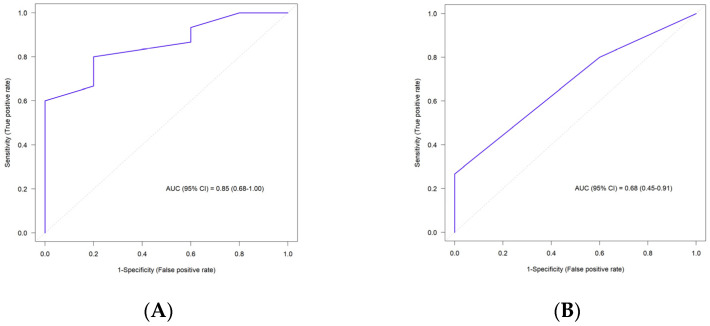
ROC plots for a model containing SNP variants for HF patients. (**A**) ROC plot for a model containing 19 SNP variants for HF patients (AUC = 0.85). (**B**) ROC plot for a model containing 4 SNP variants for HF patients (AUC = 0.68).

**Table 1 jpm-12-00744-t001:** Baseline demographic characteristics of Control group and HF patients.

Characteristic	Control Group, N = 95	HF Patients, N = 98	*p* Value
Age (years)	44.01 ± 13.8	52.7 ± 11.0	**0.001 ****
Gender			
Male	63 (66.3)	92 (93.9)	**0.001**
Female	32 (33.7)	6 (6.1)	
Ethnicity			
Asian	60 (63.2)	77 (78.6)	**0.03**
Caucasian	35 (36.8)	21 (21.4)	
Body weight (kg)	69.5 ± 14.0	79.8 ± 13.9	**0.001 ****
Height (cm)	168.3 ± 7.46	169.8 ± 6.36	0.15 *
BMI (kg/m)	24.6 ± 4.9	27.7 ± 4.5	**0.001 ****
SBP	114.4 ± 9.9	104.8 ± 15.5	**0.001 ****
DBP	75.7 ± 5.7	71.2 ± 10.3	**0.001 ****
History of smoking			
Smokers	38 (40.0)	58 (59.2)	**0.01**
Non-smokers	57 (60.0)	40 (40.8)	
Diagnosis	-		
ICM	-	44 (44.9)	
DCM	-	40 (40.8)	
HCM	-	11 (11.2)	
VHD	-	3 (3.1)	
NYHA	-		
I	-	1 (1.0)	
II	-	1 (1.0)	
III	-	2 (2.0)	
IV	-	26 (26.5)	
IIIA	-	34 (34.7)	
IIIB	-	34 (34.7)	
HF type	-		
HFrEF	-	97 (99.0)	
HFmrEF	-	1 (1.0)	
INR	-		
Basic INR	-	1.21 ± 0.36	
Target INR	-	2.39 ± 0.26	
Device strategy	-		
BTT	-	10 (10.2)	
DT	-	88 (89.8)	
Device type	-		
HW	-	18 (18.4)	
HM2	-	34 (34.7)	
HM3	-	46 (46.9)	
Warfarin dose (mg/day)	-	2.99 ± 1.15	
Duration of LVAD support till outcome, from 2011 untill 2016, *n* = 36 (in months)	-	29.6 ± 17.3	
Patients’ achieved outcome till 2017	-		
Survived	-	71 (72.4)	
Not-survived	-	27 (27.6)	
Thrombosis	-		
Yes	-	13 (13.3)	
No	-	85 (86.7)	
Bleeding	-		
Yes	-	14 (14.3)	
No	-	84 (85.7)	
Infections	-		
Yes	-	39 (39.8)	
No	-	59 (60.2)	
Stroke	-		
No Stroke	-	78 (79.6)	
Hemorrhagic stroke	-	8 (8.2)	
Ischemic stroke	-	12 (12.2)	
Myocardial infarction	-		
Yes	-	44 (44.9)	
No	-	54 (55.1)	

Continuous variables are presented, mean ± SD and categorical variables as n (%). HF patients, heart failure patients; Student’s *t*-test *p* value is labeled with one asterisk (*); Mann-Whitney U test’s *p*- value is labeled with double asterisks (**); The significant *p* value (*p* < 0.05) is labeled in bold; “-”, non-available parameters; BMI, body mass index; SBP, systolic blood pressure; DBP, diastolic blood pressure; ICM, ischemic cardiomyopathy; DCM, dilated cardiomyopathy; HCM, hypertrophic cardiomyopathy; VHD, valvular heart disease; NYHA, New York Heart Association; HFrEF, heart failure reduced ejection fraction; HFmrEF, heart failure mid-range ejection fraction; INR, International normalized ratio; BTT, bridge-to-transplantation; DT, destination therapy; HW, HeartWare HVAD; HM2, HeartMate II; HM3, HeartMate III.

**Table 2 jpm-12-00744-t002:** Comparison of baseline demographic characteristics of patients with/without complications.

Characteristic	Comparison between HF Patients		
	Group 1, N = 74	Group 2, N = 24	*p* Value
Age (years)	52.5 ± 11.3	53.4 ± 10.1	0.92 **
Gender			
Male	71 (95.9)	21 (87.5)	0.16
Female	3 (4.1)	3 (12.5)	
Ethnicity			
Asian	56 (75.7)	21 (87.5)	0.27
Caucasian	18 (24.3)	3 (12.5)	
Body weight (kg)	80.0 ± 12.2	79.3 ± 18.5	0.86 *
Height (cm)	170.0 ± 6.08	168.9 ± 7.24	0.46 *
BMI (kg/m)	27.7 ± 4.10	27.6 ± 5.66	0.97 *
SBP	105.0 ± 15.7	104.0 ± 15.0	0.99 **
DBP	70.9 ± 10.7	72.1 ± 9.35	0.33 **
History of smoking			
Smokers	46 (62.2)	12 (50.0)	0.34
Non-smokers	28 (37.8)	12 (50.0)	
Diagnosis			
ICM	36 (48.6)	8 (33.3)	0.10
DCM	25 (33.8)	15 (62.5)	
HCM	10 (13.5)	1 (4.2)	
VHD	3 (4.1)	0	
NYHA			
I	1 (1.4)	0	0.58
II	1 (1.4)	0	
III	1 (1.4)	1 (4.2)	
IV	17 (23.0)	9 (37.5)	
IIIA	27 (36.5)	7 (29.2)	
IIIB	27 (36.5)	7 (29.2)	
HF type			
HFrEF	74 (100)	23 (95.8)	0.25
HFmrEF	0	1 (4.2)	
INR			
Basic INR	1.19 ± 0.37	1.26 ± 0.33	0.11 **
Target INR	2.36 ± 0.24	2.46 ± 0.32	0.06 **
Device strategy			
BTT	6 (8.1)	4 (16.7)	0.25
DT	68 (91.9)	20 (83.3)	
Device type			
HW	11 (14.9)	7 (29.2)	**0.01**
HM2	22 (29.7)	12 (50.0)	
HM3	41 (55.4)	5 (20.8)	
Warfarin dose (mg/day)	3.01 ± 1.04	2.92 ± 1.46	0.29 **
Duration of LVAD support till outcome, from 2011 untill 2016, *n* = 36 (in months)	29.1 ± 17.6 (*n* = 21)	30.3 ± 17.5 (*n* = 15)	0.84 *
Patients’ achieved outcome till 2017			
Survived	58 (78.4)	13 (54.2)	**0.03**
Not-survived	16 (21.6)	11 (45.8)	
Thrombosis			
Yes	0	13 (54.2)	**0.0001**
No	74 (100)	11 (45.8)	
Bleeding			
Yes	0	14 (58.3)	**0.0001**
No	74 (100)	10 (41.7)	
Infections			
Yes	24 (32.4)	15 (62.5)	**0.015**
No	50 (67.6)	9 (37.5)	
Stroke			
No Stroke	60 (81.1)	18 (75.0)	0.57
Hemorrhagic stroke	5 (6.8)	3 (12.5)	
Ischemic stroke	9 (12.2)	3 (12.5)	
Myocardial infarction			
Yes	36 (48.6)	8 (33.3)	0.24
No	38 (51.4)	16 (66.7)	

Continuous variables are presented, mean ± SD and categorical variables as n (%). HF patients, heart failure patients; Group 1, without complications; Group 2, with complications; Student’s *t*-test *p* value is labeled with one asterisk (*); Mann-Whitney U test’s *p*- value is labeled is labeled with double asterisks (**); The significant *p* value (*p* < 0.05) is labeled in bold; BMI, body mass index; SBP, systolic blood pressure; DBP, diastolic blood pressure; ICM, ischemic cardiomyopathy; DCM, dilated cardiomyopathy; HCM, hypertrophic cardiomyopathy; VHD, valvular heart disease; NYHA, New York Heart Association; HFrEF, heart failure reduced ejection fraction; HFmrEF, heart failure mid-range ejection fraction; INR, International normalized ratio; BTT, bridge-to-transplantation; DT, destination therapy; HW, HeartWare HVAD; HM2, HeartMate II; HM3, HeartMate III.

**Table 3 jpm-12-00744-t003:** Comparative analysis of biochemical parameters between HF patients without complications, with thrombosis and bleeding complications.

Study Groups	Parameters	Before 14 Days	*p*-Value	After 3–6 Months	*p*-Value	After 12–18 Month	*p*-Value
Group 1	Hemoglobin, g/L	139.6 ± 18.2		129.5 ± 16.2		125.7 ± 20.0	
Group 2-1		145.8 ± 21.7	0.32 *	129.0 ± 15.3	0.93 *	125.1 ± 18.4	0.95 *
Group 2-2		128.5 ± 21.2	**0.052 ***	111.1 ± 29.2	**0.04 ****	117.6 ± 35.0	0.55 *
Group 1	Hematocrit, %	41.1 ± 6.72		38.2 ± 4.92		37.5 ± 5.46	
Group 2-1		43.1 ± 6.95	0.39 *	37.0 ± 3.67	0.51 *	36.3 ± 6.14	0.58 *
Group 2-2		39.0 ± 6.61	0.30 *	32.8 ± 8.70	**0.016 ***	35.6 ± 10.3	0.63 *
Group 1	Lymphocytes, %	27.5 ± 9.25		26.2 ± 8.79		22.0 ± 6.44	
Group 2-1		31.0 ± 8.78	0.19 **	18.4 ± 6.34	**0.015 ***	29.8 ± 11.3	**0.03 ****
Group 2-2		21.8 ± 6.28	**0.02 ****	24.8 ± 9.35	0.68 *	19.9 ± 11.3	0.27 **
Group 1	LDH, U/L	271.5 ± 137.0		273.5 ± 93.7		306.0 ± 156.1	
Group 2-1		N/A		335.4 ± 133.3	0.24 **	194.6 ± 42.7	**0.02 ****
Group 2-2		254.0 ± 171.9	0.38 **	260.2 ± 139.0	0.78 *	377.4 ± 174.2	0.19 **
Group 1	AST, U/L	27.9 ± 21.4		19.6 ± 5.50		21.2 ± 8.08	
Group 2-1		29.5 ± 10.6	0.11 **	26.9 ± 19.0	0.42 **	22.2 ± 8.44	0.77 *
Group 2-2		33.3 ± 27.0	0.43 **	36.9 ± 37.5	**0.02 ****	43.0 ± 70.3	0.63 **
Group 1	Total bilirubin, mg/dL	1.33 ± 1.56		0.70 ± 0.51		1.03 ± 1.51	
Group 2-1		1.03 ± 0.73	0.74 **	1.26 ± 0.58	**0.01 ****	0.76 ± 0.80	0.55 **
Group 2-2		3.00 ± 5.09	0.40 **	1.59 ± 2.76	0.77 **	0.96 ± 0.71	0.42 **

Continuous variables are presented, mean ± SD. Student’s *t*-test *p*-value is labeled with one asterisk (*); Mann-Whitney U test’s *p*-value is labeled with double asterisks (**); the significant *p* value (*p* < 0.05) is labeled in bold; N/A, Not available. Group 1, without complications; Group 2-1, thrombosis; Group 2-2, bleeding; LDH, lactate dehydrogenase; AST, aspartate aminotransferase.

**Table 4 jpm-12-00744-t004:** The distributions of allelic and genotype frequencies of 21 SNPs between Control group and HF patients.

Gene	SNP rs Number	Genotype	Control Group, No. (%)	Allele Frequency in Population Control Group	HF Patients, No. (%)	Allele Frequency in HF Patients	*p* Value
** *VKORC1*1* **	rs8050894	CC	17 (17.9)	C:G = 0.39:0.61	55 (56.1)	C:G = 0.68:0.32	**0.0001 ***
		CG	41 (43.2)		24 (24.5)		
		GG	37 (38.9)		19 (19.4)		
		C	75		134		
		G	115		62		
** *ITGB3* **	rs5918	TT	71 (74.7)	T:C = 0.87:0.13	48 (49.0)	T:C = 0.66:0.34	**0.0001 ***
		TC	23 (24.2)		34 (34.7)		
		CC	1 (1.1)		16 (16.3)		
		T	165		130		
		C	25		66		

Control group, healthy control group; HF patients, heart failure patients; *p* value of Hardy-Weinberg equilibrium test; *p* value (*p* < 0.05) numbers with statistical significance are labeled with asterisk (*) in bold.

**Table 5 jpm-12-00744-t005:** The distributions of allelic and genotype frequencies of 21 SNPs between HF patients with/without complications.

Gene	SNP rs Number	Genotype	Group 1, No. (%)	Allele Frequency in Group 1	Group 2, No. (%)	Allele Frequency in Group 2	*p*-Value
** *VKORC1*1* **	rs8050894	CC	41 (55.4)	C:G = 0.68:0.32	14 (58.3)	C:G = 0.71:0.29	1
		CG	18 (24.3)		6 (25.0)		
		GG	15 (20.3)		4 (16.7)		
		C	100		34		
		G	48		14		
** *VKORC1*2* **	rs9934438	GG	14 (18.9)	G:A = 0.39:0.61	0	G:A = 0.35:0.65	**0.008 ***
		GA	29 (39.2)		17 (70.8)		
		AA	31 (41.9)		7 (29.2)		
		G	57		17		
		A	91		31		
** *VKORC1*3* **	rs9923231	CC	14 (18.9)	C:T = 0.40:0.60	0	C:T = 0.35:0.65	**0.012 ***
		CT	31 (41.9)		17 (70.8)		
		TT	29 (39.2)		7 (29.2)		
		C	59		17		
		T	89		31		
** *CYP2C9*2* **	rs1799853	CC	70 (94.6)	C:T = 0.97:0.03	21 (87.5)	C:T = 0.94:0.06	0.357
		CT	4 (5.4)		3 (12.5)		
		TT	0		0		
		C	144		45		
		T	4		3		
** *CYP2C9*3* **	rs1057910	AA	68 (91.9)	A:C = 0.96:0.04	23 (95.8)	A:C = 0.98:0.02	1
		AC	6 (8.1)		1 (4.2)		
		CC	0		0		
		A	142		47		
		C	6		1		
** *CYP2C9*5* **	rs28371686	CC	74 (100)	C:G = 1.000:0.000	24 (100)	C:G = 1.000:0.000	N/A
		CG	0		0		
		GG	0		0		
		C	148		48		
		G	0		0		
** *CYP2C19*2* **	rs4244285	GG	52 (70.3)	G:A = 0.84:0.16	19 (79.2)	G:A = 0.90:0.10	0.772
		GA	20 (27.0)		5 (20.8)		
		AA	2 (2.7)		0		
		G	124		43		
		A	24		5		
** *CYP2C19*3* **	rs4986893	GG	70 (94.6)	G:A = 0.97:0.03	23 (95.8)	G:A = 0.98:0.02	1
		GA	4 (5.4)		1 (4.2)		
		AA	0		0		
		G	144		47		
		A	4		1		
** *ITGB3* **	rs5918	TT	42 (56.8)	T:C = 0.70:0.30	6 (25.0)	T:C = 0.56:0.44	**0.005 ***
		TC	19 (25.7)		15 (62.5)		
		CC	13 (17.6)		3 (12.5)		
		T	103		27		
		C	45		21		
** *GGCX* **	rs11676382	CC	70 (94.6)	C:G = 0.97:0.03	24 (100)	C:G = 1.000:0.000	0.677
		CG	3 (4.0)		0		
		GG	1 (1.4)		0		
		C	143		48		
		G	5		0		
** *CYP4F2* **	rs2108622	CC	47 (63.5)	C:T = 0.78:0.22	13 (54.2)	C:T = 0.71:0.29	0.501
		CT	22 (29.7)		8 (33.3)		
		TT	5 (6.8)		3 (12.5)		
		C	116		34		
		T	32		14		
** *UGT1A6* **	rs2070959	AA	28 (37.8)	A:G = 0.66:0.34	12 (50.0)	A:G = 0.65: 0.35	**0.03 ***
		AG	41 (55.4)		7 (29.2)		
		GG	5 (6.8)		5 (20.8)		
		A	97		31		
		G	51		17		
** *ACSM2A* **	rs1133607	CC	44 (59.5)	C:T = 0.78:0.22	15 (62.5)	C:T = 0.79:0.21	1
		CT	27 (36.5)		8 (33.3)		
		TT	3 (4.0)		1 (4.2)		
		C	115		38		
		T	33		10		
** *PTGS1* **	rs3842787	CC	72 (97.3)	C:T = 0.99:0.01	23 (95.8)	C:T = 0.98:0.02	1
		CT	2 (2.7)		1 (4.2)		
		TT	0		0		
		C	146		47		
		T	2		1		
** *F5* **	rs6025	CC	73 (98.6)	C:T = 0.99:0.01	24 (100)	C:T = 1.000:0.000	1
		CT	1 (1.4)		0		
		TT	0		0		
		C	147		48		
		T	1		0		
** *F13A1* **	rs5985	CC	53 (71.6)	C:A = 0.84:0.16	19 (79.2)	C:A = 0.90:0.10	0.879
		CA	19 (25.7)		5 (20.8)		
		AA	2 (2.7)		0%		
		C	125		43		
		A	23		5		
** *F2* **	rs1799963	GG	74 (100)	G:A = 1.000:0.000	24 (100)	G:A = 1.000:0.000	N/A
		GA	0		0		
		AA	0		0		
		G	148		48		
		A	0		0		
** *F7* **	rs6046	GG	57 (77.0)	G:A = 0.87:0.13	18 (75.0)	G:A = 0.85:0.15	1
		GA	15 (20.3)		5 (20.8)		
		AA	2 (2.7)		1 (4.2)		
		G	129		41		
		A	19		7		
** *FGB* **	rs1800790	GG	49 (66.2)	G:A = 0.82:0.18	16 (66.7)	G:A = 0.81:0.19	1
		GA	23 (31.1)		7 (29.2)		
		AA	2 (2.7)		1 (4.2)		
		G	121		39		
		A	27		9		
** *MTHFR*1* **	rs1801133	GG	37 (50.0)	G:A = 0.70:0.30	10 (41.7)	G:A = 0.65:0.35	0.742
		GA	30 (40.5)		11 (45.8)		
		AA	7 (9.5)		3 (12.5)		
		G	104		31		
		A	44		17		
** *MTHFR*2* **	rs1801131	TT	43 (58.1)	T:G = 0.76:0.24	11 (45.8)	T:G = 0.69:0.31	0.589
		TG	27 (36.5)		11 (45.8)		
		GG	4 (5.4)		2 (8.3)		
		T	113		33		
		G	35		15		

Group 1, without complications; Group 2, with complications; *p* value of Hardy-Weinberg equilibrium test; *p* value (*p* < 0.05) numbers with statistical significance are labeled with asterisk (*) in bold.

**Table 6 jpm-12-00744-t006:** Association analysis of SNP genotypes with complications’ development in HF groups (adjusted for age, BMI and gender).

Gene, SNP	Model	Genotype	Group 1, No. (%)	Group 2, No. (%)	OR (95% CI)	*p*-Value	AIC	BIC
***VKORC1* rs9934438**	Codominant	A/A	31 (41.9)	7 (29.2)	1.00	**0.0015 ***	106.1	121.6
		G/A	29 (39.2)	17 (70.8)	**2.69 (0.95–7.63)**			
		G/G	14 (18.9)	0 (0)	0.00 (0.00–NA)			
	Dominant	A/A	31 (41.9)	7 (29.2)	1.00	0.27	115.9	128.8
		G/A-G/G	43 (58.1)	17 (70.8)	1.73 (0.64–4.74)			
	Recessive	A/A-G/A	60 (81.1)	24 (100)	1.00	0.0023	107.8	120.7
		G/G	14 (18.9)	0 (0)	0.00 (0.00–NA)			
	Overdominant	A/A-G/G	45 (60.8)	7 (29.2)	1.00	**0.0057 ***	109.4	122.4
		G/A	29 (39.2)	17 (70.8)	**3.96 (1.42–11.02)**			
	Log-additive	---	---	---	0.85 (0.43–1.70)	0.65	116.9	129.8
***VKORC1* rs9923231**	Codominant	T/T	29 (39.2)	7 (29.2)	1.00	0.0024	107	122.5
		C/T	31 (41.9)	17 (70.8)	2.36 (0.83–6.70)			
		C/C	14 (18.9)	0 (0)	0.00 (0.00–NA)			
	Dominant	T/T	29 (39.2)	7 (29.2)	1.00	0.38	116.3	129.2
		C/T-C/C	45 (60.8)	17 (70.8)	1.56 (0.57–4.27)			
	Recessive	T/T-C/T	60 (81.1)	24 (100)	1.00	0.0023	107.8	120.7
		C/C	14 (18.9)	0 (0)	0.00 (0.00–NA)			
	Overdominant	T/T-C/C	43 (58.1)	7 (29.2)	1.00	**0.011 ***	110.6	123.5
		C/T	31 (41.9)	17 (70.8)	**3.55 (1.28–9.86)**			
	Log-additive	---	---	---	0.80 (0.40–1.62)	0.54	116.7	129.6
** *ITGB3* ** **rs5918**	Codominant	T/T	42 (56.8)	6 (25.0)	1.00	**0.0056 ***	108.7	124.2
		T/C	19 (25.7)	15 (62.5)	**5.37 (1.79–16.16)**			
		C/C	13 (17.6)	3 (12.5)	1.47 (0.31–7.05)			
	Dominant	T/T	42 (56.8)	6 (25.0)	1.00	**0.0079 ***	110	123
		T/C-C/C	32 (43.2)	18 (75.0)	**3.83 (1.35–10.89)**			
	Recessive	T/T-T/C	61 (82.4)	21 (87.5)	1.00	0.49	116.6	129.5
		C/C	13 (17.6)	3 (12.5)	0.62 (0.15–2.53)			
	Overdominant	T/T-C/C	55 (74.3)	9 (37.5)	1.00	**0.0014 ***	106.9	119.9
		T/C	19 (25.7)	15 (62.5)	**4.83 (1.79–13.06)**			
	Log-additive	---	---	---	1.59 (0.84–2.99)	0.15	115	128
***UGT1A6* rs2070959**	Codominant	A/A	28 (37.8)	12 (50)	1.00	0.03	112.1	127.6
		A/G	41 (55.4)	7 (29.2)	0.40 (0.13–1.17)			
		G/G	5 (6.8)	5 (20.8)	2.67 (0.59–12.07)			
	Dominant	A/A	28 (37.8)	12 (50)	1.00	0.29	116	128.9
		A/G-G/G	46 (62.2)	12 (50)	0.60 (0.23–1.56)			
	Recessive	A/A-A/G	69 (93.2)	19 (79.2)	1.00	**0.044 ***	113	125.9
		G/G	5 (6.8)	5 (20.8)	**4.40 (1.06–18.20)**			
	Overdominant	A/A-G/G	33 (44.6)	17 (70.8)	1.00	**0.02 ***	111.7	124.6
		A/G	41 (55.4)	7 (29.2)	**0.32 (0.11–0.87)**			
	Log-additive	---	---	---	1.08 (0.51–2.29)	0.84	117	130

Group 1, without complications; Group 2, with complications; AIC, Akaike Information Criterion; BIC, Bayesian Information Criterion; The *p* value (*p* < 0.05) numbers with statistical significance are labeled with asterisk (*) in bold.

**Table 7 jpm-12-00744-t007:** Summary table for MDR fit with three-way split validation (k = 3).

Level	Best Models	SNP rs Number	Training Accuracy	Classification Accuracy	Prediction Accuracy
1	*MTHFR*1*	rs1801133	69.70	68.23	50.00
2	*VKORC1*2,* *MTHFR*2*	rs9934438, rs1801131	73.79	76.04	54.17
*3	*VKORC1*2,* *UGT1A6,* *MTHFR*1*	rs9934438, rs2070959, rs1801133	88.33	82.29	95.83

‘*’ indicates overall best model.

**Table 8 jpm-12-00744-t008:** Summary table for MDR fit with three-way split validation (k = 4).

Level	Best Models	SNP rs Number	Training Accuracy	Classification Accuracy	Prediction Accuracy
1	*ITGB3*	rs5918	66.82	83.18	83.18
2	*VKORC1*2* *ITGB3*	rs9934438 rs5918	74.24	86.41	86.41
3	*ITGB3* *MTHFR*1* *FGB*	rs5918 rs1801133 rs1800790	83.79	95.16	95.16
*4	*ITGB3* *CYP4F2* *MTHFR*1* *FGB*	rs5918 rs2108622 rs1801133 rs1800790	92.12	96.77	96.77

‘*’ indicates overall best model.

## Data Availability

National Laboratory Astana (contact via phone or mail) for researchers who meet the criteria for access to confidential data. The data underlying the results presented in the study are available from the authors, phone number: +77172706501, mail: akilzhanova@nu.edu.kz.

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
