# Peer review of "Association of Genetic Polymorphisms with Complications of Implanted LVAD Devices in Patients with Congestive Heart Failure: A Kazakhstani Study"

_jpm, 2022, doi:10.3390/jpm12050744_

Round 1

Reviewer 1 Report

The manuscript submitted to the Journal of Personalized Medicine and entitled «Association of genetic polymorphisms with complications of implanted LVAD devices in patients with congestive heart failure: A Kazakhstani study» is aimed to the determination of single nucleotide polymorphisms involving to coagulation system, hemostasis function and the metabolism of the therapy of heart failure patients with left ventricular assist device and associated with the increased risk of complications. The obtained results are valuable for the modern personalized medicine, the presented manuscript is well-designed and written, a large amount of the material was analyzed, but there are some issues mainly in statistical analysis of obtained results that must be solved and explained by the authors before publication. 

  1. Gene names through the abstract, the main text, tables and supplementary materials must be highlighted in italic due to the nomenclature.
  2. What do we know about the resultsof GWAS in the context of development any complications after LVAD in the heart failure patients? Authors must summarize the recent data about genetic basis of any complications after LVAD in the heart failure patients in the Introduction section.
  3. Full inclusion and exclusion criteria must be presented in the 2.1. Study participants section.
  4. Why authors selected the SNPs listed in the 2.2. Selection of SNPs section? There are a lot of other SNPs in these genes. The full criteria of SNP selection must be presented.
  5. Line 141: RT-PCR is not correct abbreviation for real-time PCR. RT-PCR is reverse transcription PCR.
  6. SNP genotyping must be described step-by-step including chemistry (master mix, TaqMan probes, etc.), volume and concentrations of reagents and protocol of genotyping.
  7. The results of the Kolmogorov-Smirnov test or analogues (access the compliance of data with a normal distribution) must be presented to support selection of parametric statistical analysis (probably the non-parametric methods should be selected?).
  8. Considering that the authors used a lot of compared groups in their work, they must perform FDR or Bonferroni corrections to avoid type I errors in null hypothesis testing through conducting multiple comparisons.
  9. It seems to be low number of patients in the Group 2 (24 patients) for genotype-diseases associated study. Power calculations must be performed and the obtained results must be presented.
  10. Did the distribution of genotypes and alleles correspond to the population level according to 1000 Genomes or Ensemble databases?
  11. In the presented form Discussion section is very descriptive and has no critical discussion of obtained results. So, the possible pathophysiological pathways and mechanisms of obtained associations between genetic polymorphisms and other studied parameters must be discussed. It is increasing the scientific significance of obtained results.
  12. I can suggest authors to additionally perform the MDR (multifactor dimensionality reduction) analysis allowing to study gene-gene interactions and to determine the protective and risk alleles and genotypes combinations.
  13. To increase the practical value of the obtained results authors must perform the ROC analysis to access the level of significance of particular genotypes as a marker of increased complication risk in heart failure patients after LVAD.
  14. Authors need performe to more clearly statement what is the originality of the obtained results?

Author Response

Dear Reviewer!

Coauthors and I very much appreciated the encouraging, critical and constructive comments on this manuscript by the all reviewers. The comments have been very thorough and useful in improving the manuscript. We strongly believe that the comments and suggestions have increased the scientific value of revised manuscript by many folds. We have taken them fully into account in revision. We are submitting the corrected manuscript with the suggestion incorporated the manuscript. The manuscript has been revised as per the comments given by the reviewers, and our responses to all the comments are presented in the file.

On Behalf of authors,

Ainur Akilzhanova

Reviewer 2 Report

The article under review represents a study investigating the impact of single nucleotide polymorphisms on the development of complications after LVAD implantation among heart failure patients. The authors conclude that polymorphism in the genes VKORC1, ITGB3, and UGT1A6 can be used to predict LVAD complications and personalize doses of warfarin and aspirin treatment. While the study aims are sound, the following aspects should be addressed:

  • In the methods section, it is stated: “We recruited 98 HF patients (age ≥18) with implanted LVAD during 2011-2016 when they were diagnosed as BTT and DT at the NRCC.” Please indicate what type of HF the patients were suffering from, for example, HFrEF, HFmrEF, HFpEF.

  • In the statistical analysis section, please clarify if any false discovery rate correction methods were used.

  • The authors write that they have used a Student’s t-test to compare the continuous variables of two groups of patients. Has the distribution of the continuous variables been assessed?

  • Recently published studies (such as PMID: 34095631 and PMID: 34423350) have shown that machine learning models could have better capabilities of detecting risk genetic risk factors contributing to the development of heart failure and complications of antithrombotic and anticoagulant therapies. It would be beneficial to machine learning as a future perspective in the discussion section.

  • The low sample size is problematic, especially when trying to create comprehensively adjusted models.

Author Response

Dear Reviewer!

Coauthors and I very much appreciated the encouraging, critical and constructive comments on this manuscript by the all reviewers. The comments have been very thorough and useful in improving the manuscript. We strongly believe that the comments and suggestions have increased the scientific value of revised manuscript by many folds. We have taken them fully into account in revision. We are submitting the corrected manuscript with the ammendmends incorporated the manuscript. The manuscript has been revised as per the comments given by the reviewers, and our responses to all the comments are presented in the file.

On Behalf of authors,

Ainur Akilzhanova

Reviewer 3 Report

The study of Zhalbinova et al studies the impact of SNP on the outcome of LVAD implantation. This is a very relevant topic considering the detrimental outcome of follow-up therapy on the affected patients. The study is well-designed and well-presented and would be of interest to the readers.

I would like to make the following recommendations to the authors:

  1. I would recommend the authors to have the paper edited by a native English speaker to improve readability and flow of the paper. Especially the discussion part could use some polishing.
  2. Table 1: it is not always clear how to interpret the provided statistical values (e.g. for infection...). I would suggest to briefly describe the significant observations in the text and mention the specific (sub)-groups between which the significant observation was made to avoid misinterpretation by relying on the table only.

  1. Table 5: please define in the legend: AIC BIC. It would be interesting for the readers to briefly introduce the different models in the text or provide references.

  1. Line 298 – 300, please refer to table S5

  1. I would suggest changing the order of S5 and S6 due to order of appearance in the text.

  1. Although I agree that table S6 is extensive, I would recommend to somehow bringing the significant observations displayed in table S6 into the main text of the manuscript (similar to the mutual synergisms between table 5 and table S4).

Author Response

(The authors gave the same response as above.)

Reviewer 4 Report

  1. Maybe is better to separate Table 1 in two, the first as a comparision of Group1 and Group 2 with normal control, and the second, as a comparision between the Group 1 and Group 2.
  2.   In Table  2 and Table 3 present  only variables which are   significantly different among the groups. All other results can be presented  in the Supplement.  

Author Response

(The authors gave the same response as above.)

Round 2

Reviewer 1 Report

Authors improved all issues according to the previous comments.